# Impacts of Collaborative Partnership on the Performance of Cold Supply Chains of Agriculture and Foods: Literature Review

**Nguyen Thi Nha Trang** [1], **Thanh-Thuy Nguyen** [2,*], **Hong V. Pham** [3], **Thi Thu Anh Cao** [3], **Thu Huong Trinh Thi** [4] **and Javad Shahreki** [5]

1 Graduate School of Asia Pacific Studies, Ritsumeikan Asia Pacific University, Beppu 874-8577, Japan; ng21t6qc@apu.ac.jp
2 College of Business and Law, RMIT University, Melbourne, VIC 3000, Australia
3 National Institute for Science and Technology Policy and Strategy Studies, Hanoi 10000, Vietnam; phamvanhong1973@gmail.com (H.V.P.); caothuanh@gmail.com (T.T.A.C.)
4 School of Economics and International Business, Foreign Trade University, Hanoi 10000, Vietnam; ttthuhuong@ftu.edu.vn
5 Faculty of Business, Multimedia University, Cyberjaya 63100, Malaysia; javad.shahreki@mmu.edu.my
* Correspondence: thuy.nguyen21@rmit.edu.au

**Abstract:** Collaboration in a supply chain continuously proves its role in increasing the performance of supply chains, which attracts the attention of both academia and practitioners, specifically, how to generate higher impacts of collaborative partnership on the performance of supply chains and measure them. In cold supply chains of agriculture and foods, the vital need for collaboration becomes even more significant to improve the performance. Therefore, this paper reviews relevant articles derived from the Web of Science and Scopus databases. Via the Preferred Reporting Items for Systematic Reviews and Meta-Analyses (PRISMA), the research team classifies the types of collaborative partnership in cold agriculture and food supply chains, issues of the literature when analyzing collaboration impacts on the performance of CSCs of agriculture and foods, and finally, the opportunities for the future research to boost the collaboration practices in these cold chains. Following this sequence, 102 articles were eventually extracted for the systematic review to identify themes for not only addressing the review questions but also highlighting future research opportunities for both development of partnership integration and performance of the cold chains of agriculture and foods.

**Keywords:** supply chain partnership; supply chain collaboration; supply chain performance; cold supply chain; agriculture and foods

## 1. Introduction

Cold supply chains (CSCs), or in another words named "cold chains (CCs)", are playing an important role in the global trade. Cold chains have been widely applied to diverse industries such as fresh agricultural produce, aquaculture production, frozen food, dairy products, flowers, chemicals and pharmaceutical products [1–3]. Several conceptualizations of CSCs or CCs have been introduced in the existent literature, from simple to complex ones. Castiaux [4] defines cold chain succinctly as the supply and distribution chain for products that must be stored within a specific temperature range. Consolidating the concept of CCs, Mallik et al. [5] consider CC as "a continuous and cohesive process" where all materials, equipment, procedures and labor are utilized to provide the temperatures between +2 °C and +8 °C for preserving commodities and products while in transit throughout the distribution and storage phases, from the point of origin to the point of consumption. Kitinoja [6] refers it to the uninterrupted handling of temperature-sensitive food products within a low temperature environment during the postharvest steps of the value chain. Because the products are moved and managed through various handoffs in the chain including harvest, collection, packing, processing, storage,

transport and marketing, until reaching the end-customers, any break incurred among these phases can signify a disruption in CC management, causing food safety dilemmas.

As with any supply chain, a CC adds value to customers. A well-maintained CC is expected to enhance values to customers such as to prolong the product life and facilitate the market timing which enhances value of the chain [7]. A CC creates an ideal ambient temperature around perishables and other similar products to preserve them in a safe, wholesome, and secure manner from production through to consumption [8]. Therefore, there have been many studies examining how to increase the performance of CSCs, for instance, Parreño-Marchante et al. [9] illustrate the important role of cold stores when processing fish to cooled temperature room and then distributing to the market in the aquatic supply chains, or Bonou et al. [10] recognizes the potential contribution of super-chilling to supply chain augmentation for exporting products in the pork supply chain, to name as a few.

Among many approaches to improve the supply chain performance, supply chain partnership is the one which have long been assessed as critical to consolidate supply chains [11]. Fugate and Mentzer [12] have summarized the benefits of supply chain coordi-nation (or cooperation) from the literature: risk reduction, access to resources, competitive advantage, lower costs, higher profits and value improvement, etc. As a result, firms have increasingly put their efforts into improving partnerships within their supply chain [13] so that their supply chains can become more stable and sustainable through the avoidance of risks and the accumulation of benefits derived from partnerships [14,15].

Collaborative partnership mechanisms in supply chains may vary depending upon the approaches adopted by scholars to classify the interdependencies between actors or activities for each kind of supply chain network. The collaborative partnership in supply chains is essential to ensure their performance and thus sustainability. The cooperation is along the supply chain by virtue of the coordination between various members within the chain [16]. Therefore, numerous studies on how the collaboration influences the cold supply chain performance have been conducted. The topic has become even more prevalent since the outbreak of the COVID-19 pandemic, such as with Pérez-Mesa et al. [17] who examine the collaborative relationships for sustainability in the agrifood supply chain, Papaioannou et al. [18] who analyse the role of adverse economic environment and human capital on collaboration within agri-food supply chains, or Aggrey et al. [19] who discuss the firm performance implications of supply chain integration in agri-businesses, etc.

Collaborative partnership, therefore, is an emerging actor toward cold chain perfor-mance. However, there is a variety of challenges that cold chain's partners have been facing when establishing the collaboration, which requires classifying the anticipated benefits and approaches for sharing the benefits and cost as well as requisite investments among supply chain members [12–14]. Lambert et al. [15] provide a list of challenges when working on partnership facilitation, such as scheduling difficulties, prioritizing action items, ensuring the allocation of managerial resources, etc. Mofokeng et al. [11] confirm that the collabora-tion can fail due to the partners' reluctance to share information, apply effort and bring in investment. In addition, the cooperation can be effected by the negotiation power among partners in the supply chain [16].

Given the crucial meaning of collaboration in cold supply chains, it is thus important to investigate the types and contexts of the collaborative partnerships impacting on the cold supply chain performance and how these contribute to their sustainability. However, due to the many different types of CC, of which CCs of agriculture and foods products play a significant role, this paper, therefore, only presents a literature review on the topic focusing on the CCs of agriculture and foods (CCAFs).

The remainder of the paper is presented as follows: the next section introduces the review methodology, followed by the theoretical foundation of collaborative partnership and supply chain's performance. After that is the descriptive statistic of the literature. The subsequent section illustrates the thematic analysis and presents the main themes

and elaboration of the gaps in the literature. Finally, the directions and future research opportunities are summarized, and the conclusions conclude the study.

## 2. Review Methodology

To identify various types of collaborative partnership in CCAFs, dimensions of CCAF performance, and their causal relationships, this study adopted a systematic review of the literature on the CSCs. In contrast to the traditional or narrative review, systematic reviews provide a more rigorous and well-organized approach to review the literature in a specific issue [20]. In this regard, while a conventional narrative review summarizes articles to elucidate a generally broad topic without any scientific method, a systematic review involves a critical and reproducible synthesis of the high quality, available publications on the same topic, helping to avoid biases in research [21]. To gain an in-depth understanding of how collaborative partnerships affect the performance of CC and CCAF, a systematic review is considered as well suited to this study by virtue of its extensive ranges of context, business and operation strategies.

This research applies the procedure of systematic review under PRISMA (Preferred Reporting Items for Systematic Reviews and Meta-Analysis) approach to screen the dataset originally proposed by Linares-Espinós et al. [21], which was then tailored to fit the context of CCAFs. Although this method is time-consuming and labor intensive, it has been proved to generate high-quality results in many comprehensive literature review papers [22]. To further elaborate on the objective of this literature review, two review questions were employed:

RQ1: What are collaborative partnerships employed by stakeholders in CSCs?
RQ2: How do collaborative partnerships impact on the performance of CSCs?

To answer these RQs, a systematic review is necessarily implemented to examine all forms of collaborative partnership available in a cold chain and their impacts on various perspectives of cold chain performance.

### 2.1. Criteria Identification

#### 2.1.1. Scope of the Literature to Be Reviewed

This study chooses to review the literature located at the intersection of collaborative partnerships in CCAFs and their performance. The authors initially examine a broad literature of supply chain to comprehend theoretical foundations of the study. Since we focus on the collaborative partnership in the context of CCAFs' performance, we first summarize the concepts of collaboration/collaborative partnership in supply chains and CSCs from the approaches of the supply chain's stakeholders. Following the literature, collaboration is often interchanged with several words, namely integration, coordination, and cooperation, depending on the situations of application or disciplines [23]. Accordingly, the literature of CSCs is composed of both general CCs and supply chain. Furthermore, collaborative partnerships in supply chain can be interchanged with alternative words related to coordination, cooperation, or integration.

The review process is restricted to English papers which is considered as the globally established language of scientific communication and also to fit the authors' foreign language proficiency [21].

#### 2.1.2. Type and Timeframe of Publication

For the sake of rigor, this review paper investigates only peer-reviewed published journal articles and excludes dissertations, textbooks, conference papers, letters and notes, etc. Although the concept of supply chain management was introduced at the beginning of the 1980s, research in this field has been almost scarce and only become exponential since the mid-1990s [24]. Thus, it is reasonable to select the time period from 1990 to 2021 for searching and examining articles in this study.

### 2.2. Data Acquisition

A characteristic to distinguish a systematic review from a traditional narrative one is an extensive literature search, which is essential to avoid omitting any potential studies with which really relevant ones may be mingled [21]. Therefore, when sourcing the data, we chose to accept low precision at the beginning to attain high sensitivity later. Specifically, we initiated the process of data acquisition by identifying data sources and relevant keywords to construct search terms around supply chain collaboration. After all articles related to supply chain collaboration were screened, we would narrow down the data to the ones only involved in cold chain and then the impacts of collaboration on cold chain performance [21].

In order to have a good coverage of curated scholarly documents, the evidence was first retrieved from Scopus and Web of Science digital libraries, known as two world-leading databases that are increasingly used in academic papers [25]. Based on the initial examination of supply chain literature as previously explained, two sets of keywords were used to detect preliminary relevant studies out of existing data. The first set regarding supply chain includes *supply chain*, *value chain*, and *chain*. The second set relating to collaboration contains *collaboration*, *coordination*, *cooperation*, *integration* and *partnership*. We also consider different word classes of these keywords by using asterisks (*), combined with the usage of the English terms "OR" and "AND", to search documents in the databases. For instance, the search term "collaborat*" would be entered to find *collaboration*, *collaborate*, *collaborated*, *collaborative*, and *collaborating*. A string of keywords was created afterward to source the publications, namely *("Supply chain" OR "Value chain" OR "Chain") AND ("Collaborat*" OR "Coordinat*" OR "Cooperat*" OR "Integrat*" OR "Partnership")*. Finally, aiming at a focused and accurate search, we allowed detecting the articles whose titles only, not abstracts or keywords, contained the search terms.

The criteria were also applied to the first data collection, whereby the search was limited to the journal articles in English published from 1990 to present. The search returned a total of 6970 documents from Scopus and 4626 documents from Web of Science where Web of Science Core Collection was selected. Since the focus of literature review is put on cold chain in agriculture and foods, the totally irrelevant subject areas were left out, such as engineering electrical electronic, physics applied, nanotechnology, oncology, mathematics, thermodynamics, political science, materials science and biology, etc. The second refinement released an aggregate of 7541 papers, in which 4821 papers from Scopus and 2720 papers from Web of Science. Next, the authors used Excel tool to remove the duplicated items, reducing the data size to 5321 papers.

### 2.3. Data Screening and Synthesis

In the stage of data screening, the title and abstract of each paper were thoroughly read to judge whether the papers discussed about collaboration in a particular cold chain or a supply chain of certain product type requiring cold storage. A list of 118 papers was afterward synthesized for full text examination so that only those mentioning impacts of collaborative partnerships on performance of cold chain would be selected. This stage led to a final number of 102 papers obtained for data analysis and interpretation. Some papers, whose titles involved in typical supply chains likely to be cold chains as agri-food chain, were still excluded since the specific supply chain described in their content did not demand for low temperature preservation. Figure 1 presents the data screening and synthesis process conducted in this study.

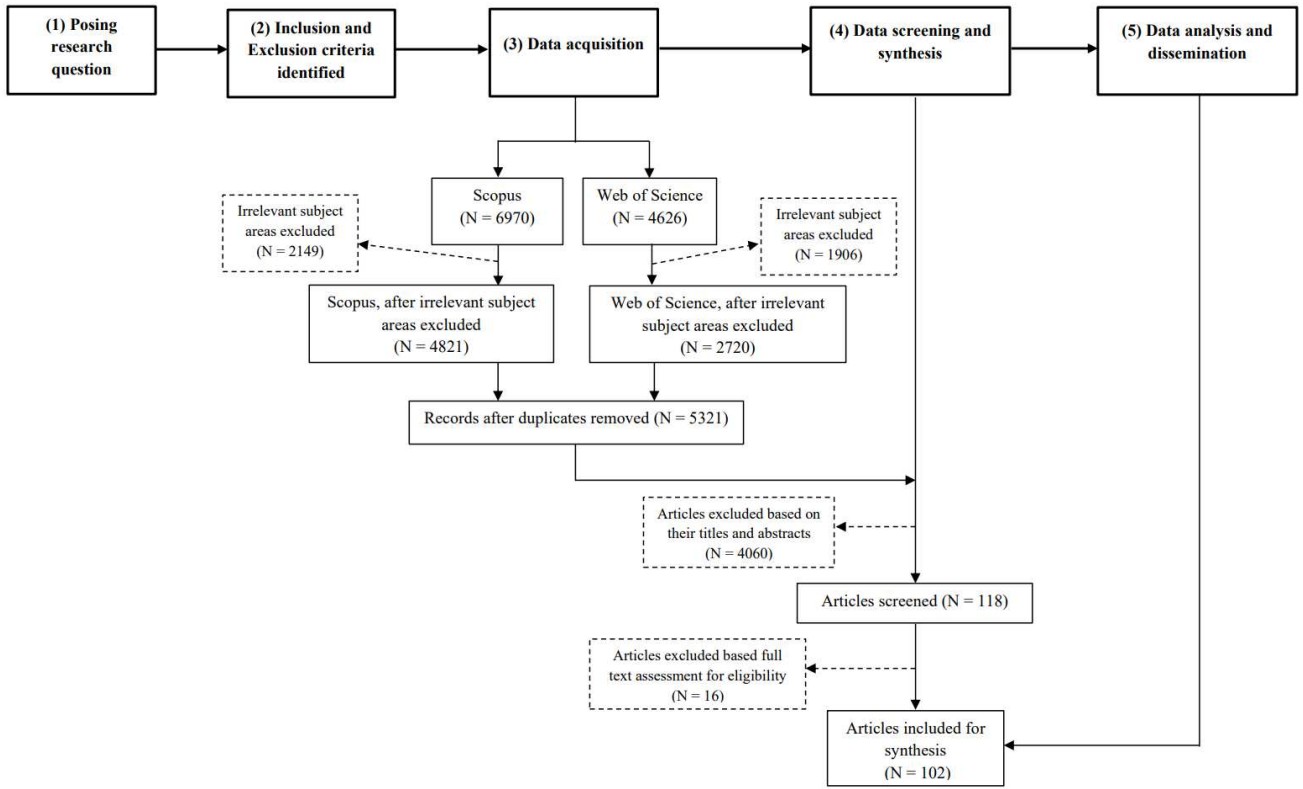

**Figure 1.** The procedure of systematic review under PRISMA approach.

*2.4. Data Analysis and Dissemination*

Both descriptive and thematic analysis were carried out in this study with respect to the data analysis. While descriptive analysis quantitatively summarizes basic features of selected articles, thematic analysis allows for an iterative process to make a map of the most important themes or patterns within the data.

**3. Theoretical Foundation**

*3.1. Collaborative Partnerships in Supply Chains*

In the context of supply chain, the word *collaboration* is often interchanged with several words, namely *integration*, *coordination*, and *cooperation*, depending on the situations of application or disciplines [23]. Originated from two Latin word elements, "col" and "laborare", the term "collaboration" means "to work together" [26]. From an academic aspect, "collaboration" is understood as "a process of joint decision making among key stakeholders of a problem domain about the future of that domain" [27]. For any context of research, "collaboration" is mainly used to infer individuals or organizations working together towards a common goal, together with mutual benefits, shared rewards and discussed risk information as its foundation [28]. Similar to "collaboration", "coordination" is the essence of supply chain management [12], whereby the chain stakeholders may behave as a part of a unified system and coordinate with each other [29]. Finally, supply chain collaboration is determined by the extent to which internal functions of a business and its supply chain parties strategically cooperate with each other to jointly administer intra- and inter-organizational processes [30]. Integration is regarded as a competitive strategy [31], which enables a supply chain to overcome another if this integration really works effectively. As a result, when tracing the relevant literature of collaborative partnerships in CSCs in agriculture and foods, the research team also considers other interchangeable terms, including coordination, integration and cooperation.

Types of collaboration in SCs can be divided by scope of collaboration, be decision function, by spectrum of relationship, and by shared components.

### 3.1.1. By Scope of Collaboration

Based on the scopes of collaboration and specific parties tending to collaborate with each other in supply chain, Mark [28] divides supply chain collaboration into two major categories: vertical and horizontal. The vertical form refers to relationships with suppliers, customers and internally across functions. This kind of collaboration can also be interpreted as supply–demand, supplier–producer, manufacturer–customer and logistics provider collaboration [32]. The horizontal form includes collaboration internally across functions, and with competitors and non-competitors who are at the same level of the chain to share resources such as warehouse space and manufacturing capacity [33]. Business functional processes internally collaborated may include purchasing, manufacturing, logistics and marketing [34,35]. Meanwhile, external interfaces that a firm can co-work with its exterior partners embrace collaborative planning, forecasting and replenishment (CPFR).

### 3.1.2. By Decision Function

According to Malone [36], the so-called decision function (meaning how supply chain partners decide what actions to take) is an attribute bringing about two collaboration styles: centralized and decentralized. In the centralized decision form, there exists a single firm responsible for primary control of other firms belonging to the chain; whereas in the decentralized form, individual firms make their decisions autonomously [37,38]. Following this trend, many researchers have set up and solved complex optimization problems of supply chain in both scenarios of centralized and decentralized system.

### 3.1.3. By Spectrum of Relationship

There are some authors who use a spectrum of relationship types between organizations to determine different levels of collaboration in supply chain, ranging from Arm's length to Partnerships, Joint ventures, and Vertical integration [39].

Arm's length relationship is purely transactional [40], whereby parties have no sense of joint commitment or joint operations. This transactional relationship is different from partnerships in which parties build up mutual trust and share openness, risk and reward to establish a competitive advantage and yield greater business performance than that which would be achieved by the firms individually.

Meanwhile, joint venture involves shared ownership across two companies and vertical integration entails the merger of ones that are in the same industry but undertake different stages of supply chain: production or distribution [39].

Finally, vertical integration is introduced by Cooper et al. [41] as a new form of supply chain collaboration based on multiple paths to supply chain integration named "dyadic relationship". In this form, an organization attempts to coordinate and communicate with its immediate channel members (channel integrators), whereby a channel leading company sets the overall strategy and get other members committed to the channel strategy [42].

### 3.1.4. By Shared Components

Finally, there are several studies defining supply chain collaboration through the activities and processes which are willingly shared by stakeholders within the chain, named "components". These activities and processes are created and controlled by managers throughout the life of a partnership to make the relationship more operational and beneficial to all parties. For instance, Gardner et al. [39] define that the elements of collaboration consist of joint planning, joint operating controls, communication, risk/reward sharing, trust and commitment, contract styles, scope and financial investment. Later, Min et al. [43] discuss the culture of sharing as the nature of collaboration and suggest similar shared activities and processes among collaborative partners in the supply chain. They are planning, problem solving, performance measurement, resources and skills, goals, objectives, benefits, risks and information.

However, owing to different perceptions and/or aims, different authors may conceptualize the same components of collaboration in different ways. For instance, Simatupang and

Sridharan [44] introduce "Incentive alignment" to imply shared costs, risks and benefits; and "Decision synchronization" to infer joint long-term planning and processes of order generation and delivery. Similarly, Cao and Zhang [45] synthesize seven interconnecting components: information sharing, goal congruence, decision synchronization, incentive alignment, resources sharing, collaborative communication, and joint knowledge creation. Table 1 presents the types of collaboration in supply chains.

**Table 1.** Types of collaboration in supply chains.

| Types of Collaborative Partnerships in Supply Chains | | | | |
|---|---|---|---|---|
| **By Scope** | **By Decision Function** | **By Spectrum of Relationship** | **By Shared Components** | |
| Vertical | Centralized | Arm's length | Incentive alignment | Information sharing |
| | | Partnerships | | Goal congruence |
| Horizontal | Decentralized | Joint ventures | Decision synchronization | Decision synchronization |
| | | | | Incentive alignment |
| | | Vertical integration | | Resources sharing |
| | | (dyadic relationship) | | Collaborative communication |
| | | | | Joint knowledge creation |

### 3.2. Performance Measures in Supply Chains

To characterize performance of a supply chain, there is a bank of measures, and each has its own pros and cons [46]. Two conventional groups of performance measures are profit-oriented and client-oriented. The former traditionally includes various types of costs [47]: inventory costs and operating costs. Reduced costs are also discussed by Skjoett-Larsen et al. [48] as a benefit of supply chain collaboration. Some other common financial measures are added subsequently, namely, total cost of resources used, distribution costs, manufacturing cost, inventory holding cost, Return on Investment (ROI), sales, profit by Beamon [47], profitability by Min et al. [43] and Attaran and Attaran [49], and cash to cash cycle by Barber [50]. Moreover, client-oriented performance measures range from fill rate, on-time deliveries, stock-out probability, customer response time, lead time, shipping errors, customer complaints to flexibility [47], then supplemented with customer perceived value of product by Gunasekaran et al. [51] and Elrod et al. [46]; reliability by Bhagwat and Sharma [52]; and product quality in the forms of percentages of product accepted at source inspection and of damaged goods on arrival by Barber [50]. There are some authors who emphasize the importance of a balance between cost cutting and customer satisfaction [46], or more broadly, financial and nonfinancial measures [53]. Moreover, companies are now facing increased pressure from customers and governments regarding their environmental and social responsibilities [54,55]. Hence, environmental sustainability has become a crucial dimension of an effective supply chain and an assessment of environmental performance should be embedded in the performance system besides conventional profit- and customer-oriented performance [55].

Ultimately, since a cold chain possesses many special features that distinguish it from other types of supply chain such as short shelf life, refrigerated transportation and storage requirements, traceability, product appearance, taste, colour and size, seasonality in production, etc., it is quite challenging to measure the performance of a cold chain [2]. Besides similar performance measures to a primary supply chain, some innovative performance factors are added to the performance system of cold chain such as the continuity of food supply [56], improvement in cost control and innovation [57], stability of perishable goods in logistics chains [58], and traceability [59–61]. Both prior and up-to-date performance measures of CSCs thus classified into three overarching performance dimensions: profit-oriented, client-oriented and environment-oriented. Figure 2 provides an overview of classification scheme for supply chain collaboration and (cold) supply chain performance retrieved from the literature.

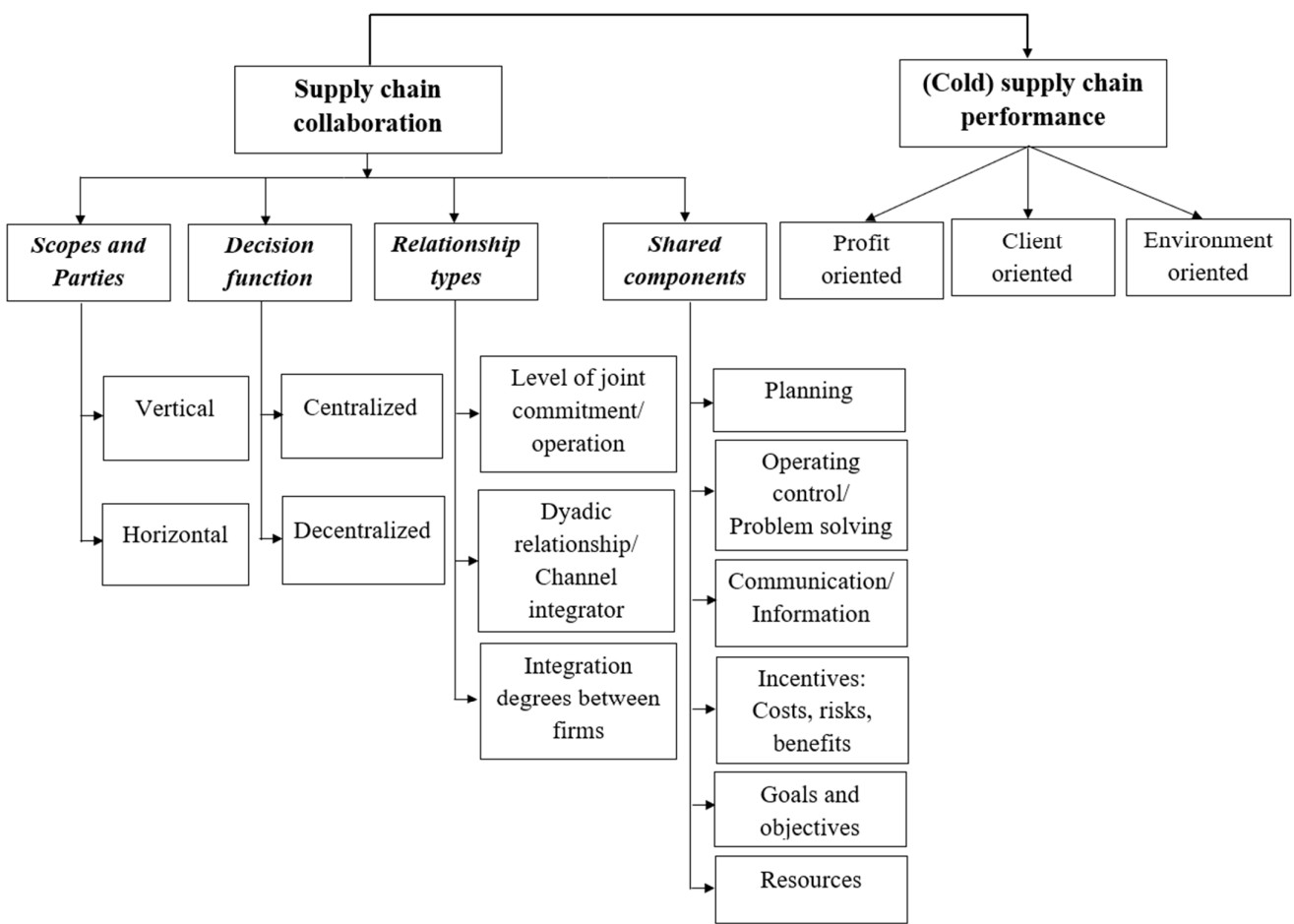

**Figure 2.** Classification scheme for supply chain collaboration and CSCs' performance.

## 4. Descriptive Statistics

In this section, we analyse the general information of the selected 102 articles following the distribution of articles by time, by the Scimago ranked journals, the top journals with the highest number of articles in review, the geographic region, and finally the research area (types of CSCs under the review).

### 4.1. The Distribution of the Articles by Time

Figure 3 presents the number of relevant papers published from 1990 to 2022. Obviously, this number is quite limited during 1990–2007 with only one or two articles annually or even no articles. However, there is an upward trend exhibited since 2008 although the growth of papers is sporadic throughout the years. This trend implies an increasing interest in studying the impacts of collaborative partnerships on cold chain outcomes. The year 2020 witnessed the highest number of publications with 18 papers recorded, whereas a decrease was observed in the following year, 2021. The sampling process came to an end in April 2022, leading to the whole of this year not being covered.

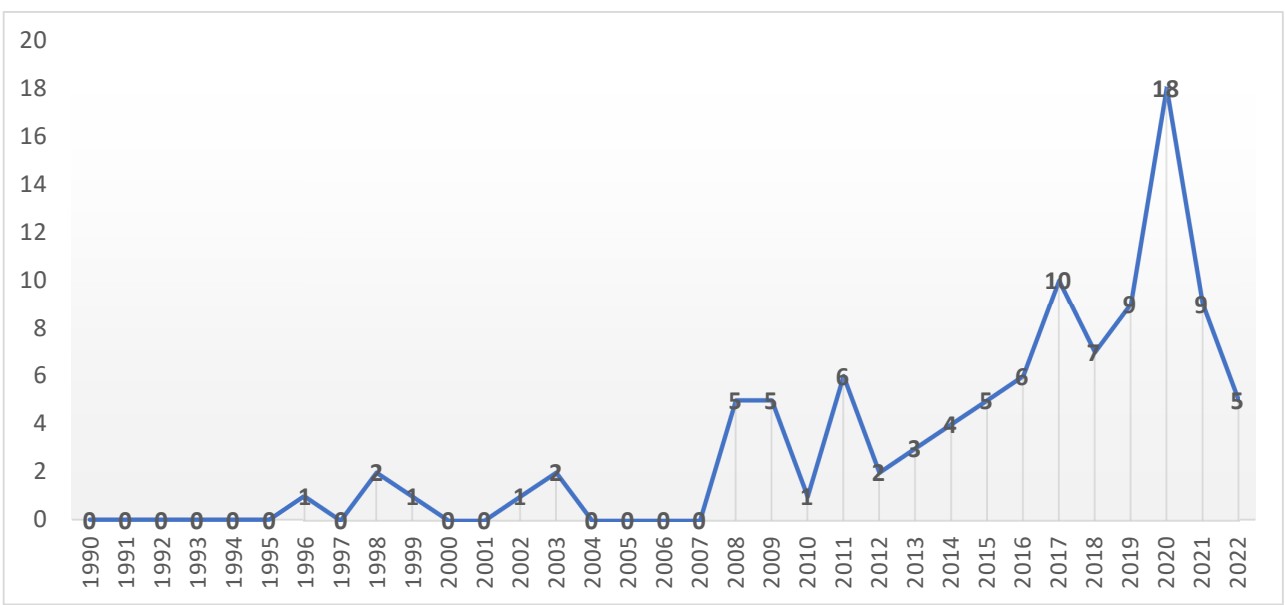

**Figure 3.** Total number of articles published by year.

*4.2. The Distribution of the Articles by Scimago Ranked Journals*

The selected papers were published in a total of 64 journals. According to Figure 4, most of the selected articles are published in high-quality journals found on the Scientific Journal Ranking (Scimago) website in 2020, with 47% and 30% of them ranked as Q1 and Q2 journals, respectively.

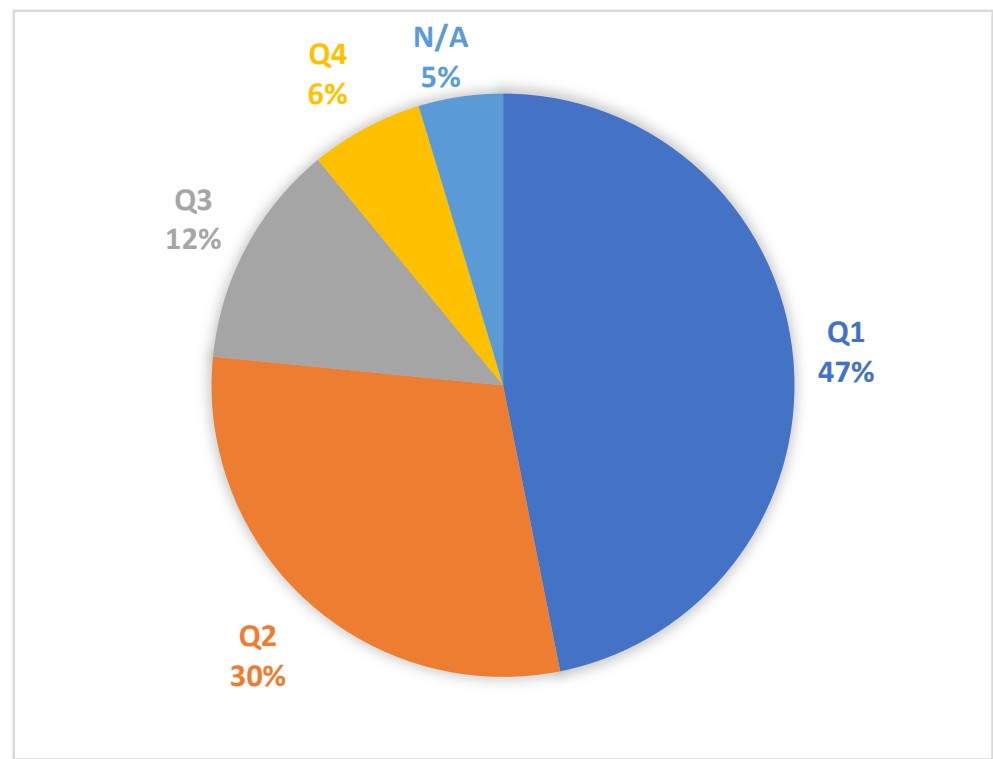

**Figure 4.** Distribution of reviewed articles in Scimago ranked journals.

### 4.3. The Distribution of the Articles by Top Journals

Among 64 journals, there are the 10 journals in which papers in our review are most frequently published. Even though these initial ten journals only account for 15.6% of total journals, they significantly contribute 40.2% to the total number of papers reviewed, as shown in Figure 5. The remaining journals devote only one or two articles to the whole collection.

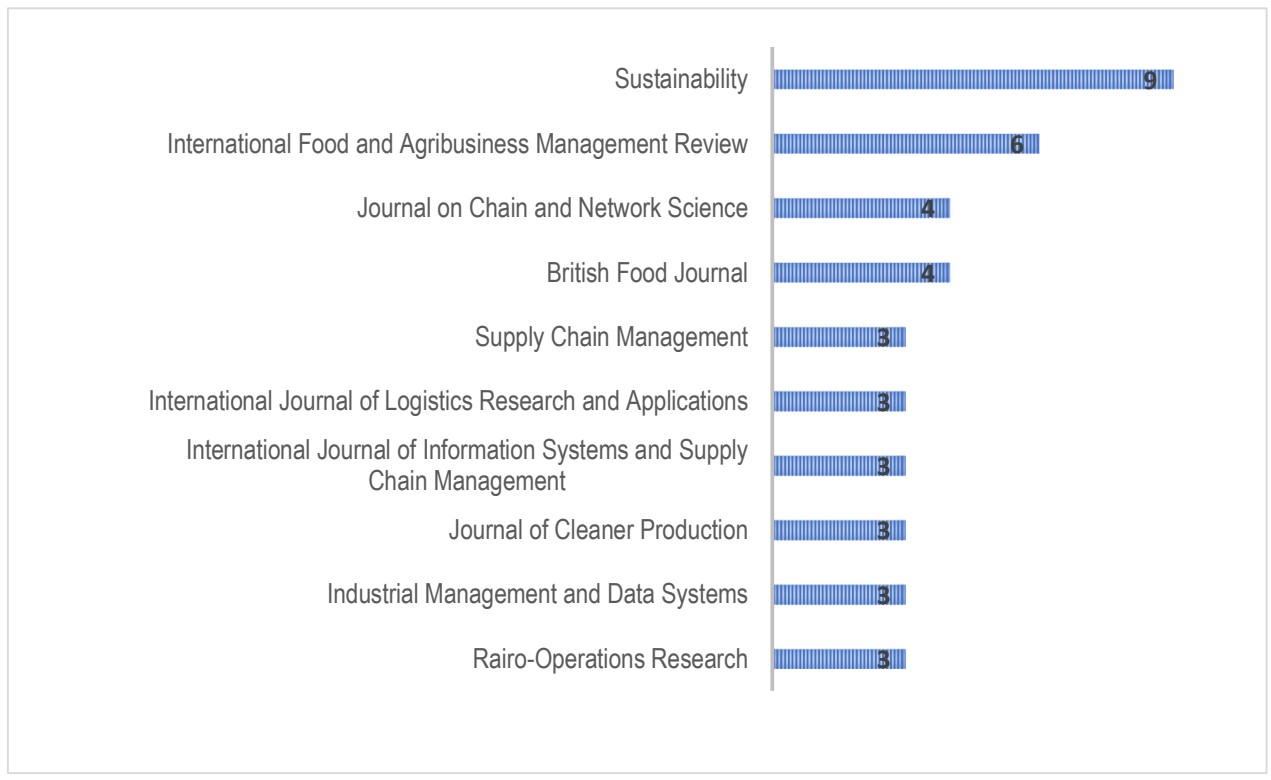

**Figure 5.** Top 10 journals with the highest number of articles in review.

The top leading journal in the field providing the largest source of the selected articles is *Sustainability* (nine articles, account for 9% of the total articles), followed by *International Food and Agribusiness Management Review* (six articles, equivalent to 6%). Both *Journal on Chain and Network Science* and *British Food Journal* have four articles (4%), while the last six journals (*Supply Chain Management*, *International Journal of Logistics Research and Applications*, *International Journal of Information Systems and Supply Chain Management*, *Journal of Cleaner Production*, *Industrial Management and Data Systems*, and *Rairo-Operations Research*) have only three articles each (3%).

### 4.4. The Distribution of the Articles by Research Region

Figure 6 illustrates the list of geographic regions in which specific countries were discussed as the research context in the reviewed articles. Totally 27 countries crossing five continents are under examination, while 29 articles (29%) did not mention any specific region. Among those, most frequent studies were conducted in three continents in sequence, named Asia, Europe and Africa with 25 (25%), 24 (24%) and 14 (14%) papers found, respectively. Among Asian countries, China attracts the greatest attention of authors with 12 articles recorded (12%), which approximately equals the number of articles discussing Africa (14%).

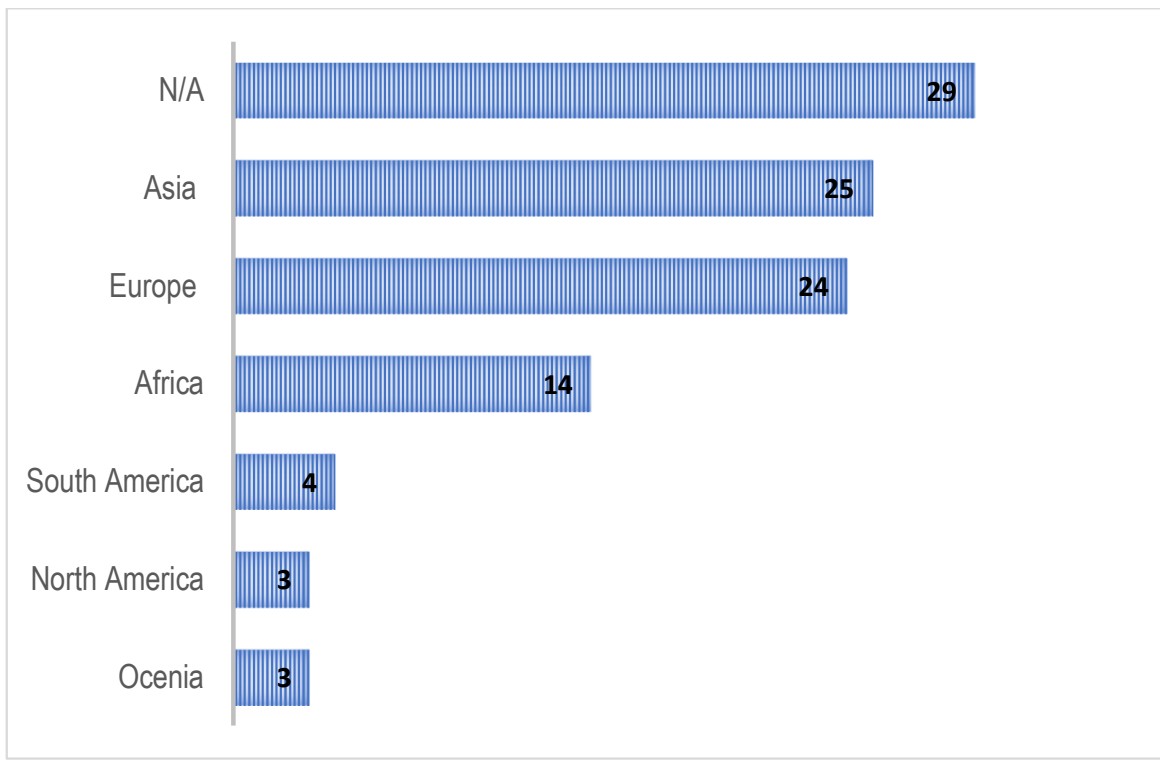

**Figure 6.** Top number of articles by research region.

*4.5. The Distribution of the Articles by Types of CCAFs*

The total 102 reviewed papers are spread out in four types of CCAFs: agriculture CSCs, cold food SCs, cold meat SCs, cold dairy SCs, and CSCs of aquamarine and fishery. Among those, the highest number of articles examines agriculture CSCs (50 articles out of total 102 articles, equivalent to approximately 49%), following by general cold food SCs (27 articles or 27%), cold meat SC (12 articles or 12%), diary supply chain (9 articles or 9%), and finally CSC in aquamarine and fishery (4 articles or 4%).

The significant number of publications on agricultural CSCs shows the vital role of this supply chain in the global economy explaining the high attention of the authors. This is quite understandable since farm products are highly diverse, ranging from vegetables, fruits, flowers to all other horticulture products [20]. They are also particularly sensitive to high temperatures. For instance, in the Near East and North Africa, the percentage of fruits and vegetables suffering losses and wastes due to the lack of sufficient and efficient cold chain infrastructure is the highest (accounts for 55%), followed by meats, fish and seafood, and dairy [62]. Among those studies, 33 papers (32.4%) examine general agri-food chains as the most popular supply chains under examination. Meanwhile, more specific products including vegetables (e.g, tomato and potato), fruits (e.g, apple, avocado, and peach), or grains are only found in one or two papers. Figure 7 presents the distribution of the articles by types of cold supply chain.

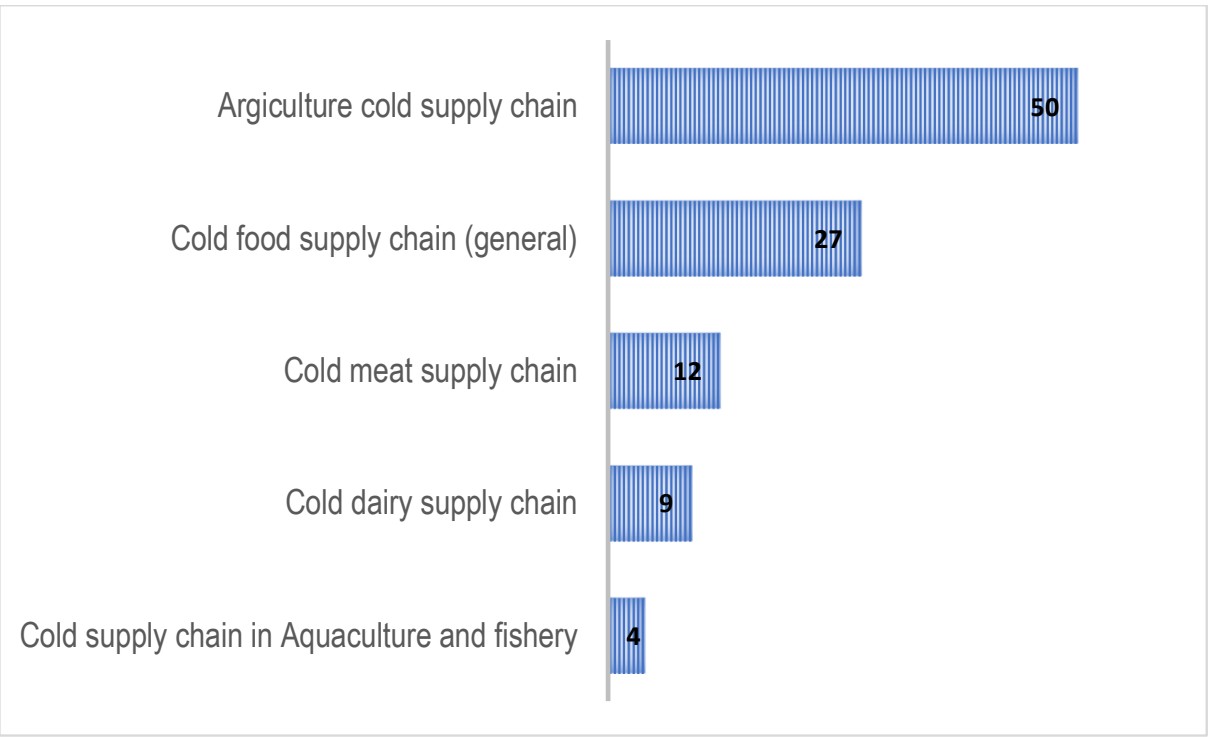

**Figure 7.** The distribution of the articles by types of cold supply chain.

## 5. Thematic Analysis of Reviewed Literature

In this step, we review each article in detail to understand how partners are coordinating in CCAFs. We also examine the impacts of the coordination on the CCAFs' performance. Following this sequence, the section therefore has three sub-sections, named collaboration in CCAFs, impacts of collaborative partnership on the performance of CCAFs, and challenges and issues of collaboration in CCAFs.

### 5.1. Collaboration in CCAFs

Collaborative partnerships in CCAFs can be evaluated by types of collaboration and types of collaborative partners.

By Types of Collaboration

Following the theoretical foundation, the collaborations in CCAFs are analysed following the collaboration types: Scopes, Decision functions, Spectrum of relationships, and Shared components. The data presented in Table 2 demonstrates that the preferred collaboration types in CCAFs are collaboration by scopes and shared component (32 and 31 papers, or 31.4% and 30.4%, respectively), while the collaboration types by relationships and decision functions are less favoured (23 and 16 papers, or 22.5% and 15.7%, respectively). Specifically, looking more closely at each collaborative type of CCAFs, the collaboration by scopes and by shared components are more favourable in CSCs in Agriculture, Foods, and Meat, while the collaboration by relationship and components is preferred in the dairy cold chains. Finally, the collaboration by shared components is strongly favoured in the CSCs in aquamarine and fishery.

**Table 2.** Collaboration in CCAF by types.

| No | Types | Agriculture | Foods | Meat | Dairy | Aquamarine and Fishery | Total |
|----|-------|-------------|-------|------|-------|------------------------|-------|
| 1 | Scopes | 18 | 8 | 2 | 2 | 1 | 31 |
| 2 | Decision functions | 8 | 5 | 2 | 1 | 0 | 16 |
| 3 | Relationship | 11 | 6 | 3 | 3 | 0 | 23 |
| 4 | Shared component | 13 | 8 | 5 | 3 | 3 | 32 |
| 5 | **Total** | **50** | **27** | **12** | **9** | **4** | **102** |

### 5.2. By Types of Collaborative Partners

There are several types of collaborative partners relating to the cooperation in the analysed CCAFs. If divided by the supply chain process, we have collaboration among internal partners and collaboration between internal and external partners.

The internal partners in the reviewed CCAFs include farmers/growers, suppliers, logistics/3PLs, brokers, wholesalers, processors/manufacturers/producers, distributers/retailors, buyers/consumers, exporters, and importers. The external partners are governments, government authorities, NGOs, associations, investors/capital owners, and other external actors (Bankers, insurers), etc. The list of papers discussing the collaborative partners is presented in Table 3.

**Table 3.** Collaborative partners in CCAFs.

| No | CCAFs | Collaborative Partners | References |
|----|-------|------------------------|------------|
| 1 | Agriculture CSCs chains | Internal stakeholders | [63–101] |
| | | Int. and Ext. stakeholder | [63,67,71,76,102–107] |
| 2 | Cold food supply chains | Internal stakeholders | [17,18,92,108–129] |
| | | Int. and Ext. stakeholder | [129,130] |
| 3 | Cold meat supply chains | Internal stakeholders | [19,57,117,120,131–138] |
| 4 | Cold dairy supply chains | Internal stakeholders | [78,139–144] |
| | | Int. and Ext. stakeholder | [144,145] |
| 5 | CSCs in aquamarine and fishery | Internal stakeholder | [71,146–148] |

The most significant number of studies (88 articles, account for 86.3%) focus on investigating the collaborative partnership among internal stakeholders of the CCAFs. To name a few, Hobbs et al. [132] examine the vertical scope of partnerships between Integrators–Contracting Firms–Retailers–Consumers to enhance the client satisfaction (trust). Under the same scope, Sharma and Patil [127], Kalykova et al. [74], and Imami et al. [143] explore the collaboration among Farmer–Processor, Processor–Retailer, Wholesaler–Exporter, and Farmers/Primary Producer–Transporter–Logistics Provider–Processor–Distributor–Consumer. Hardman et al. [149] examine the spectrum of relationship and point out that when improving the collaboration among Producers–Packers–Exporters, the production planning, delivery scheduling and quality control of the CSCs will be improved. Amanor-Boadu et al. [86] prove that enlightening the participants' commitment level and responsibilities will promote the performance and efficiency of all the partners in the studied CSCs (Farmers–Producers–Retailers–Restaurants–Wholesalers–Consumers–Processors). Cai et al. [96] analyse the cooperation of producer–distributors as a centralized collaboration. Zhao and Wu [128], and Guohua [95] investigate the behaviour of Suppliers–Producers–Retailers, and Supplier–Retailer. Yang et al. [65] discussed the alterations among Suppliers–Distributers–Retailers in the shade of the collaboratively shared component. Mishra and Dey [76] examine the decentralized model using multiple actors involved in

agricultural value chains to decide the level of integration among actors, while Yang and Peng [150] and Yang and Yao [93] analyse the cooperation of Supplier–Retailer.

Additionally, discussing the internal relationship but horizontally, the collaboration between Farmer–Farmer is argued in [99], or the evaluation of IoT collaboration among all partners in the CSC [76,77], or resource sharing among internal partners within the CCAFs [78,79].

There are only 14 papers (13.7%) debating the cooperation among internal and external actors. For instance, Kalykova et al. [74], Badraoui et al. [121], and Widadie et al. [151] deliberate the connection between agri-organizations with external bodies such as government and investors. This insignificant number shows the intemperance of the research area; in addition, there is no study exploring this topic in the cold meat supply chains and CSCs in aquamarine and fishery, leaving this topic as potential for future research.

### 5.3. Impacts of Collaborative Partnership on the Performance of CCAFs

The literature has discussed significantly the impacts of the collaborative partnership on the performance of CCAFs, from several perspectives. The collaborative partnerships are proved to create positive impacts on performance of all the five CCAFs in the literature in both internal and external collaboration. Because there is no single study discussing only client-oriented and environment-oriented as mentioned in the theoretical content, the impacts on the performance of CCAFs have been ranged following dimensions: (i) profit-oriented, (ii) profit and client-oriented, (iii) profit, client, and environment-oriented, and (iv) Client–environment-oriented. Table 2 presents the findings following the five CCAFs.

(i)     Profit-oriented dimension

The first group of impact contains the studies focusing on optimal production planning outcomes (e.g, optimal fresh-keeping effort level of the supplier, product's quality control, etc.), optimal costs and cost–relevance (e.g, reducing all types of costs, shared costs, costs traceability, etc.), optimal wholesale price, retail price, and price mechanism of different collaborating partners within the CSC. There have been totally 51 articles examining this dimension from a number of viewpoints, accounting for 50% of the total number of articles.

Representatives of this trend include Naspetti et al. [98] who examine the cost shared for highly integrated supply chains and high perceived risk for quality and safety among Producers–Packers–Processors–Traders, Hu and Xu [79] and Liu [91] illustrate the revenue and cost sharing contract for decentralized agricultural product supply chain coordination between producers and processors. Both of Ma et al. [81] and Song et al. [83] explore the centralized and decentralized models to control the cost shared in fresh agriculture chains. However, Ma et al. [81] focus on the collaboration among Supplier–3PL–Retailer, while Song et al. [83] analyse the cooperation among Producer–3P–Customers. As illustrated in Table 2, this dimension is discussed mainly in agriculture CSCs, Cold Meat Supply Chain, and CSCs in Aquamarine and fishery. Table 4 discusses the impacts of collaborative partnership on performance of CCAFs.

**Table 4.** Impacts of collaborations on performance of CCAFs.

| Types of CSCS | Drivers of Collaboration | Impacts on Performance of CCAFs | Performance Dimensions | References |
|---|---|---|---|---|
| Agriculture CSCs | Optimising SC's resources and costs | Reduction in transaction costs, logistics costs, perishability, wastage, risk, price, selling cycle length, | Profit-oriented | [81–83,92,118] |
| | Optimising supply chain network | Increase productivity, product's quality control; Optimal network design of CCs. | | [74,99,127] |
| | Optimising production planning, wholesale price, retail price, supplier's price, quantity retail price, retail quantity | Optimal production planning outcomes, Optimal wholesale price, and retail price; Optimal fresh-keeping effort level of the supplier; | | [65,70,75,77,91,93,96,150] |
| | Improving efficiency/effectiveness of the CCs (Co-invest in Tech and process, add more value, etc.) | Ensure demand security, nonseasonal availability assurance; | | [85,87,97,99,143] |
| | Sharing revenue, cost, resources through contracts | Cost reduced, cost traceability, revenue increased | | [69,88,89,94,95,100,101,128] |
| | Improving the vertical integration, partners commitment, responsibilities | More optimal production planning, delivery scheduling and quality control. | Profit and client-oriented | [86,104,122,134,149] |
| | Developing IS among partners, Sharing information and benefit, Implementing information traceability, Generating mutual trust and communication | Enhance trans-shipment strategies, traceability, partner's credit facility. Increase level of integration in CCAFs Increase stakeholders' resilience | | [63,66,68,72,79,84,90,98,107,112,121,152] |
| | Leadership collaboration | Improve performance through collaborate strategic plan | | [78,110,124] |
| | Strengthening Public–Private partnership mechanisms | More flexibility and efficiency for CCAFs, attract investment; stable development. | | [71,103,104,106,120,153] |
| | Improving client satisfaction Enhancing product's quality and specification | Reduction in delivery time, increase customer's demand and satisfaction | Client and environment-oriented | [132,154] |
| | Applying safety standards | Greener CCAFs | | [104,151] |

Table 4. *Cont.*

| Types of CSCS | Drivers of Collaboration | Impacts on Performance of CCAFs | Performance Dimensions | References |
|---|---|---|---|---|
| Cold Food Supply chains | Improving the product quality, | CCAF has higher proportion of high-quality products | Profit-oriented | [119] |
| | Improving efficiency and effectiveness in supply chains, more efficient contracts | More efficient exchange of information and organizational structures; Improve market position of the CCAF; Implied positive impacts can be achieved through vertical coordination mechanism | | [112,117,123] |
| | Implementing the quality management and premium price; Corporate social responsibility | Implied positive impacts on performance through raw material traceability and standard compliance | | [115,119] |
| | Incorporating collaboration, supply–demand synchronization, traceability and vertical integration; | Optimal warehousing and location design, supply–demand synchronization, checking price variations, minimizing waste, improving productivity | | [127] |
| | Farmers—Farmers to achieve economies of scale | Increasing profit and customer demand | | [114] |
| | Innovation implemented Developing vertical relationships | | | [17,108,125] |
| | Transaction collaborated | Reduce the transaction cost, | | [109] |
| | Coordinating agri-food chain with perishable good | Revenue sharing contract for the CCAF | | [128] |
| | Developing intersectoral partnerships | Promote sustainable change from a governance and a development perspective | | [129] |
| | Leadership collaboration | Increasing coordination effectiveness | | [110] |
| | Developing a competition and cooperation models for pricing | Maximising the total profit | | [92] |
| | Collaboration in wastage treatment Increase the chain efficiency | Reduce the food waste, wastage, | Profit, client, and environment-oriented | [111,113] |
| | Improving partnership, trust, confident, commitment; information exchanged | Obtained traceability, strategic partnership, sustainability performance for the CCAFs, | | [116,121,124,126,129] |
| | Enhancing collaborative strategies; improving types of collaboration | Implied positive impacts on performance | | [18,122] |

**Table 4.** *Cont.*

| Types of CSCS | Drivers of Collaboration | Impacts on Performance of CCAFs | Performance Dimensions | References |
|---|---|---|---|---|
| Cold Food Supply chains | Reduce tax rates or subsidy rates for suppliers and manufacturers | Reduce carbon tax, preservation technology T-subsidy mechanism; Achieve three social capital mechanisms | Client and environment-oriented | [118,130] |
| Cold Meat Supply Chain | Facilitating sustainable development through collaboration; Share profits and risk; Exchange relationship | Partners coordinate efficiently and effectively, reduce transaction cost, Optimal price mechanisms, volume, quality, resources allocation; Changing consumer demand; | Profit-oriented | [120,132,154,155] |
| | Build IT-based value chains | Share cost, share margin | | [131] |
| | Meeting product specifications. | Supply-end and market-end vertically integrated chains | | [19,133] |
| | Coordination mechanism improving information sharing | Improve resilience of the CCAF to the uncertainty of the market; Improve quality management strategies; More responsive to the evolution of the sustainability initiative | Profit and client-oriented | [134,136] |
| | Institutional innovations for more competitive, collaborate in several levels | | | [135,137] |
| Cold Dairy Supply Chain | Getting to higher market | Higher standards of CCAFs | Profit and client-oriented | [140] |
| | Providing more choices of coordination mechanisms in the presence of transaction costs | Optimal location of producer, source of market information, distance to markets, travel time to buyers or suppliers, etc. | | [139] |
| | Resolving challenges that limit smallholders' integration in value chains | Implied positive impacts on performance of CCAFs | | [141] |
| | Promoting communication, trust, common understanding, knowledge exchanged | | | [138,142,143,145] |
| | Collaborate with government agencies for innovation | Implied positive impacts on performance of CCAFs | | [144] |
| | Enhancing managerial competencies and capabilities of dairy farm managers | Increase control and efficiency along the supply chain | | [78] |

**Table 4.** *Cont.*

| Types of CSCS | Drivers of Collaboration | Impacts on Performance of CCAFs | Performance Dimensions | References |
|---|---|---|---|---|
| CSCs in Aquamarine and Fishery | Developing virtual corporation platforms | Implied positive impacts on performance of CCAFs | Profit-oriented | [146] |
| | Developing a coordination model for aquatic supply chains | Manage the quality risk of aquatic products | | [71] |
| | Improving management of the CSCs | Impacts on performance though boosting coordination in production and inventory management | | [147] |
| | Collaborating as an attribute of Sustainable SCM | Lean management drive economic benefits for the CCAF. | | [148] |

(ii)    Profit and client-oriented dimension

The second sub-group includes 39 articles (38.2%) analysing the impacts on CCAFs' performance through supply chain's strategic integration, information sharing, compliance, trusts, and so on. For instance, Akhtar and Khan [110] demonstrate that performance can be improved through leadership collaboration between supplier–importer. Daniel et al. [84] analyse the relationship between Farmer–Buyer to enhance the trust, relational satisfaction, symmetric power dependency, and relational commitment. Bitzer et al. [104] raise the solutions to improve the position of smallholder farmers and their organizations and increase the efficiency of producer's assets by using the intersectoral partnerships among large multinational companies–NGOs–onsite implementer–technology provider. As mentioned in Table 2, the outcomes of CCAFs can be promoted through sustainable change through collaboration between governance and other partners, developing intersectoral partnerships and leadership collaboration, enhancing information sharing and communication, stakeholders' resilience, etc. This dimension is widely implemented in cold meat supply chain and cold dairy supply chain.

(iii)    Profit, client, and environment-oriented dimension

The third group of articles discuss several similar above issues under the lens of all three dimensions with only nine articles in cold food supply chains, accounting only for 9% of the total publications. For instance, collaboration in wastage treatment [88,89], increasing the chain efficiency, improving partnership–trust–confident–commitment–information exchanged [78,90–93] enhancing collaborative strategies, and improving types of collaboration [18,94], etc.

(iv)    Client and environment-oriented dimension

The final group of literature analyses environmental issues on applying safety standards for greener CCAFs [151], and reducing carbon tax, preservation technology T-subsidy mechanism, achieving three social capital mechanisms [94,95]. Totally only three papers (accounting for 3% of the total articles) have been conducted on this field, proposing the potential area for the future research.

## 6. Discussion and Opportunities for Future Studies

Based on the outcomes of the systematic review of the literature in the domain, several themes can be drawn relating to the collaborative partnership in the agriculture CSCs and cold foods supply chains.

Firstly, the study's finding indicates that research on CCAFs has currently focused on collaboration in profit-oriented and profit and client-oriented dimensions to create impacts on CCAFs' performance. There have been so many advantages that collaboration can create for boosting the performance of CCAFs, for instance, in the profit-oriented dimension, there are reduction in cost (transaction costs, logistics costs, cost traceability), reduction in perishability, wastage, risk, price, selling cycle length, while increasing productivity, product's quality control, as well as solutions for optimal network design, optimal production planning outcomes, optimal wholesale price and retail price, etc. In the profit and client-oriented dimension, collaboration helps enhancing trans-shipment strategies, traceability, partner's credit facility, increasing stakeholders' resilience, more flexibility and efficiency for CCAFs, at the same time attracting investment and guaranteeing a stable development, etc.

However, in the sense of using technology to serve the profit-oriented dimension, there is a lack of studies discussing the application of the new technologies and information systems supporting the collaboration, which helps increase the performance of the CCAFs, such as the efficiency of the logistics system, the product quality and quantity, and reduce the food waste and wastage, etc. In addition, since fragmenting happens in farms and market which results in high lead time, cost, waste, order return, customers' complaints and dissatisfaction [111,113], high technologies in farms can give support to prolong product life and shared-information technologies can closely estimate the customer demand for

farmers, which in turn manages more significant producing plans with higher quality of products and price for customers.

The Industrial Revolution 4.0 has accelerated the widespread application of digital technologies (DTs) in various sectors and supply chains recently [156–158]. For instance, Marinelli et al. [159] prove that an IoT (Internet of Things) platform is beneficial for all partners in cold-chain logistics while providing higher efficiency levels of CSCs and increasing the customers' satisfaction. Other studies by Zhang et al. [160] and Tsang et al. [156] confirm that implemented blockchain and IoT-based technology under the background of e-commerce cold-chain logistics can significantly improve the quality and safety of the CSCs, specifically in aquatic products [160]. Pérez-Mesa et al. [17] suggest that the incorporation of new technologies implemented in processes within the cold supply chains will require a rethinking and redesign of the whole of the supply chain business model and cooperation. Innovative technology such as big data, Internet of Things (IoT), artificial intelligence, blockchain, and so forth, can be applied to transform and facilitate collaboration in the framework of an intelligent supply chain, making them aspects that require further research [17]. In conclusion, it will be beneficial for all stakeholders to embed DTs in the subsiding ecosystem of the CCAFs, which supports the identification, measurement and evaluation of information and resource sharing, collaborating strategy, and collaborative planning between various partners [112,118]. Thus, the embeddedness of digital technologies and technical support systems in promoting collaboration in CCAFs is becoming more and more critical while the number of papers arguing this topic is still limited. This gap in literature needs to be filled by future works.

Secondly, with 50% of the studies reviewed profit-oriented dimension, and 38.2% of the literature analysed profit and client-oriented dimension (i.e., 51 and 39 papers out of 102 papers, respectively) using mainly economical metrices as tools to manage the impacts of coordination on CCAFs' performances, it is confirmed that economical optimal solutions have been considered as an efficient method to maintain and enhance performance through collaborative partnership. Moreover, the impacts under the client-oriented dimension have been discussed quite thoroughly through the uncertainty of the market, the quality management strategies, rate of response to the evolution of the sustainability initiative [134–137], and optimal location of producer, source of market information, distance to markets, travel time to buyers or suppliers [139,140]. However, there have been quite a few studies considering the collaboration with customers for more social responsibilities and ethical business to improve the sustainability of the whole supply chains.

Thirdly, there has been an extremely limited number of studies discussing the impacts on performance under the environment-oriented dimension or relevant to environmental-oriented dimension, except only three papers suggesting collaborative partnership for supporting safety standards and reducing carbon tax to upsurge the CCAFs' performance [118,130,151]. This positions a necessity for further research on the topic. Specifically, if separately analysing each type of CCAFs, the literature of cold meat supply chains, cold dairy supply chains, and CSCs in aquamarine and fishery does not have any studies on collaboration partnership conducted under the environment-oriented dimension, leaving this gap opening wide for the future studies. Specifically, nowadays the mitigation in foot waste, wastage, energy consumption, use of toxic materials, or promoting recycling, use of eco-labeling, green packaging, and eco-friendly operation in CCAFs have become essential for businesses and their supply chains rather than just a choice. Therefore, research on how collaboration can create positive impacts on CCAFs' performance to achieve these objectives for cold meat supply chains, cold dairy supply chains, and CSCs in aquamarine and fishery will be highly welcomed.

Fourthly, given the complex nature of the agriculture and foods cold supply chains in which multiple players and systems are mutually interrelated, an extra dimension of collaboration with a stakeholder would lead to domino effects on the others. Hence, managing collaboration among partners would require collaborative inputs from other

relevant primary and secondary stakeholders. In addition, the measures designed to manage performance should be oriented not only internally within the supply chain internal partners, but also external stakeholders, i.e., relevant regulatory bodies, local government, associational agents, banks, insurance, etc. However, while there is an enormous number of studies (88 articles, account for 86.3%) examining the collaborative partnership among internal stakeholders of the CCAFs, there are only few articles exploring the cooperation between internal and external stakeholders and its impacts on the performance. This also positions ongoing studies in the future.

Finally, measures managing collaboration are derived from well-known and widely acknowledged principles of "quality management", "business continuity management" and "sustainability management" across the literature. To maintain the sustainability of the CCAFs, all collaborating partners would need to devise the strategic plan of collaboration to identify and analyse the most suitable form of coordination for their organization, evaluate the advantages and disadvantages as well as prepare well for business continuity, and employ environmental measures such as recycle contingency and eco-friendly operation planning. Meanwhile, each stakeholder may need to be flexible in adopting lean principles to be more efficient in the collaborative operating environment, precisely in the context of the modern supply chains.

Given the above discussion, the opportunities for future research are summarised accordingly (Figure 8). From the collaboration's holistic level, because CSCs contain a variety number of stakeholders depending on the reviewing perspective, the future research in any possibility mentioned must clearly establish and understand the context, which is the nature of the CSCs and the collaborating environments where the collaborative partnership originates from. In addition, both internal and external stakeholders of the CSCs and their interdependent relationships in the supply chain setting are also identified and classified accordingly. For each stakeholder's interest and their own decision-making process and strategy, the value-added of under reviewed collaboration must be properly identified, analysed and assessed to conclude its positive impacts of the performance. Based on the outcomes of this assessment, collaborating strategies will be devised accordingly, which can create positive impacts to the performance of the CSCs.

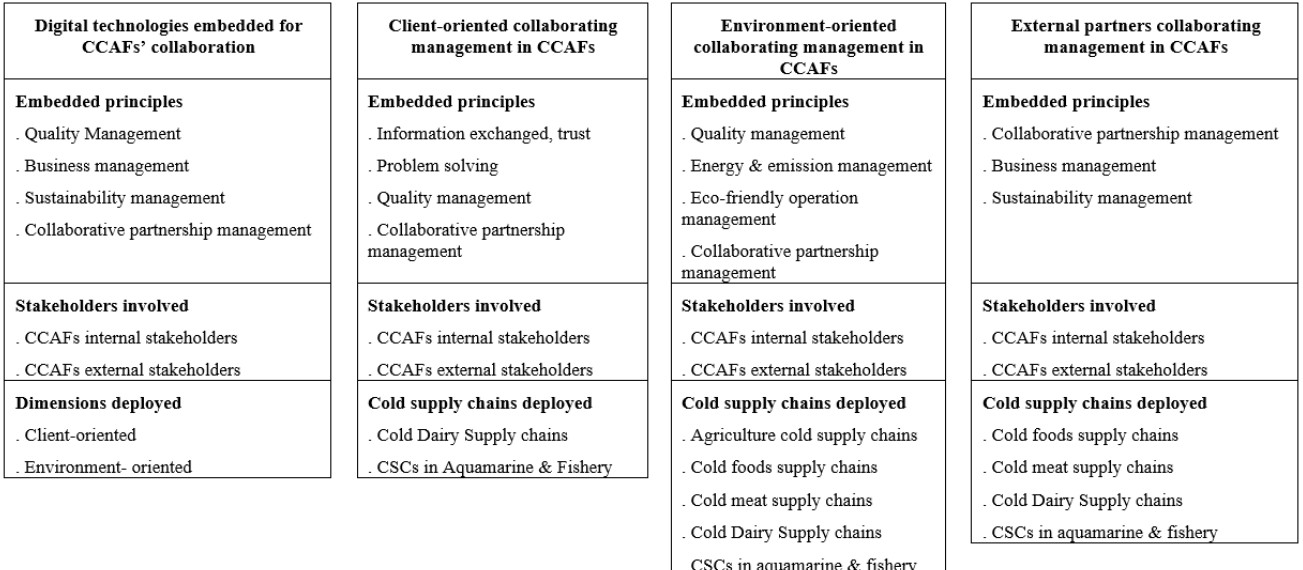

**Figure 8.** Opportunities for future research.

## 7. Conclusions

The CCAFs have an indispensable role in global trade. They are complex with multiple stakeholders, systems, and networks, which are mutually interrelated. In this context, managing collaboration in the CCAFs is critical, as any interaction of one stakeholder

in the chain would create momentous impacts on others and the chain's overall performance. This paper presents a systematic review of the literature in this critical domain. The findings show that a significant number of earlier studies mainly focused on economic measures under the profit-oriented dimension. These studies are designed using mathematical models and management frameworks to make decisions based on optimal SC's resources and costs, supply chain network, production planning, wholesale price, retail price, supplier's price, quantity retail price, retail quantity, etc., to analyse and evaluate the impacts of the collaboration on the chain's partners. There is, however, a lack of research empirically examining the embedded digital technologies for supporting the ecosystem of the CCAFs. Other issues can be listed here, such as the lack of studies examining (1) the collaboration with customers for adding social responsibilities and ethical business to improve the sustainability of the whole supply chain, or (2) how these collaborations could contribute to the performance of CCAFs under client-oriented and environmental-oriented dimensions; or (3) only a few articles exploring the cooperation between internal and external stakeholders and its impacts on the performance. Moreover, several challenges observed when establishing and maintaining cooperation are also argued and discussed, however, only in a few of the articles.

Future research opportunities, therefore, are put forward, in which four paths of research questions can be proposed, as shown in Figure 8. These research opportunities can add value to both theory building and management practice if taken well. Acknowledging that this review was conducted only on the two prestigious databases, i.e., Web of Science and Scopus, future research may need to expand the review to other databases to cover all the literature. Such studies in the future will enhance knowledge building in the CCAFs and supply chain management overall while providing meaningful theoretical and managerial implications to both academics and professionals working on the topics of CCAFs.

## 8. Limitations

The study contains several limitations which impact on the findings. First of all, the analysis could be in a greater detail if the research team analysed the variety factors measuring each SC's performance, and not only focus on three dimensions of SC's performance. Secondly, the review was conducted only on the two prestigious databases, i.e., Web of Science and Scopus, which means publications in other databases may not be collected in this review. Another limitation comes from the keywords search following the title and abstract of the papers, which means a limited number of papers discussing the collaboration in the content only could be missed here. Considering the complex issues of collaboration in CSCs, the study can be extended by using more keywords reflecting all types of the collaboration, which may yield a larger number of papers.

**Author Contributions:** Introduction, conceptualization: N.T.N.T. and T.-T.N.; Methodology: N.T.N.T., T.-T.N. and H.V.P.; Theoretical background: N.T.N.T., T.T.A.C. and T.H.T.T.; Discussion: T.-T.N. and J.S.; Writing—original draft preparation: N.T.N.T., T.-T.N., H.V.P., T.T.A.C., T.H.T.T. and J.S.; Writing—review and editing: N.T.N.T. and T.-T.N.; Project administration: T.T.A.C. and H.V.P. All authors have read and agreed to the published version of the manuscript.

**Funding:** This research was funded by the National Foundation for Science and Technology, Vietnam (Nafosted).

**Institutional Review Board Statement:** Not applicable.

**Informed Consent Statement:** Not applicable.

**Data Availability Statement:** Not applicable.

**Acknowledgments:** We are grateful for the Nafosted (National Foundation for Science and Technology, Vietnam).

**Conflicts of Interest:** The authors declare no conflict of interest.

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
