# Peer review of "Impacts of Collaborative Partnership on the Performance of Cold Supply Chains of Agriculture and Foods: Literature Review"

_sustainability, doi:10.3390/su14116462_

Round 1
Reviewer 1 Report
The topic undertaken by the authors is very important. It is a comprehensive and systematic literature review. The authors described the research procedure in detail.
The conducted analysis of the literature allowed the authors to identify significant research gaps, of which the key elements include corporate social responsibility. The proposed paths for further research should be positively assessed. According to the authors, carrying them out may allow for the development of the theory and practice of food supply chain management.
Author Response
Dear Reviewer 1,
Thank you for your comments. The research team thoroughly revised the most important content of the manuscript, in which the proposed paths for further research are analysed in "6. Discusisons and opportunities for future studies" and Figure 8.
Reviewer 2 Report
The research paper is good.
Author Response
Thank you for your evaluation.
Reviewer 3 Report
General comment:
This paper reviews relevant articles (about 102 articles) derived from the Web of Science and Scopus databases. This manuscript aimed to identify the impacts of collaborative partnership on the performance of cold supply chains of agriculture and foods. The manuscript is good and can be accepted after addressing the following appended points:
- Line 48-49: Please improve the clarity of this sentence
- Line 94-95: incomplete sentence
- Line 113: please number the subtitles
- Line 138-145: please add a reference to these procedures
- Line161-162: Why did the authors select the journal articles in English published from 1990 to the present?
- A diagrammatic figure should be provided to outline the procedures of this study.
- Line 194-272, section 3.1 Collaborative partnerships in supply chains should be summarized in a table
- Line 305-308: This figure is not clear. Please upload another one.
- Line 320-321: Figure 3 should be adjusted well, especially for the horizontal axis. Otherwise, using a table would be better.
- Line 341: there were two figures with the same terminology as figure 3 (please see also line 305-308) please adjust this issue.
- Line 344-359: Please insert the P-value for the significance.
- Line 360-371: The same comment as in line 344-359. Also, please address the same comment for the section in line 378-391
- Line 425: is it significant % (I am talking about: a high proportion of studies (88 articles, account for 86.3%)).
- Line 444-446: please combine this part with the next paragraph in line 447-452.
- Table 3: please confirm the uniformity of the reference style in this table and the text.
- From page 23 to the end, no lines to follow. Please adjust this point.
- Please add some description for the figure 7
- Please summarize the conclusion section. More importantly, I encourage the authors to create a limitation section for this study.
Author Response
Many thanks to reviewers for your comments on the manuscript.
Below are our responses to the comments. All the changes in the revision are used “track changes” and marked in blue colour.
